# Motivators and Influencers of Adolescent Girls’ Decision Making Regarding Contraceptive Use in Four Districts of Zambia

**DOI:** 10.3390/ijerph20043614

**Published:** 2023-02-17

**Authors:** Mumbi Chola, Khumbulani W. Hlongwana, Themba G. Ginindza

**Affiliations:** 1Discipline of Public Health Medicine, School of Nursing and Public Health, University of KwaZulu-Natal, Durban 4041, South Africa; 2Department of Epidemiology & Biostatistics, School of Public Health, University of Zambia, Lusaka 10101, Zambia; 3Cancer & Infectious Diseases Epidemiology Research Unit (CIDERU), College of Health Sciences, University of KwaZulu-Natal, Durban 4041, South Africa

**Keywords:** adolescent girls, contraceptive use, motivators, influencers, decision making, Zambia

## Abstract

Low contraceptive use in sub-Saharan Africa, and Zambia specifically, negates the potential benefits of contraception in preventing unwanted and early pregnancies. This study aimed to explore and understand the motivators and influencers of adolescent girls’ contraceptive decision making. Using thematic analysis, we analysed qualitative data from seven focus group discussions and three key informant interviews with adolescent girls aged 15 to 19 years in four Zambian districts. The data were managed and organised using NVivo version 12 pro (QSR International). Fear of pregnancy, fear of diseases, fear of having more children, and spacing of children (especially among married adolescents) were key motivators for adolescents’ contraceptive use. Friends and peers motivated them to use contraceptives while fear of side effects and fear of infertility drove non-use. Peer pressure and fear of mocking by their friends were important deterrents to contraceptive use. Parents, peers and friends, family members, partners, churches, and religious groups influenced adolescent girls’ contraceptive decisions. Mixed messages from these influencers, with some in favour and others against contraceptives, make adolescents’ decisions to use contraceptives complex. Therefore, interventions targeting increased contraceptive use should be all-inclusive, incorporating multiple influencers, including at institutional and policy levels, to empower adolescents and give them autonomy to make contraceptive decisions.

## 1. Introduction

Globally, adolescent sexual and reproductive health has been prioritized in recent years [1] due to the significant death, illness, and injury that adolescents face, most of which can be prevented or treated [2]. Universal access to sexual and reproductive health (SRH) services and rights by 2030 is included in the Sustainable Development Goals (SDGs), numbers 3 and 5, under which adolescents are included [1]. Adolescents comprise a significant proportion of the population, with a global estimate of 1.2 billion, which is about one-sixth of the world population, the greatest number of adolescents ever recorded [2]. This is expected to increase further by 2050, especially in low- and middle-income countries where nearly 90% of adolescents reside [2]. Adolescents face various health challenges, which may be age specific, such as water, hygiene, and sanitation-related risks for younger adolescents (10–14 years), as well as behavioural (alcohol use and unsafe sex) risks for older adolescents (15–19 year olds) [2]. Globally, among adolescent girls aged 15–19 years, pregnancy complications and unsafe abortions are the leading causes of death [2]. Annually, in low-income countries, approximately 21 million adolescent girls between the ages of 15 and 19 years become pregnant, while approximately 12 million give birth and about 777,000 births are reported among adolescent girls aged below 15 years [3].

In Zambia, contraceptive use among adolescents remains low while adolescent girls become sexually active at an early age. Statistics show that the median age at first sexual intercourse among women aged 25–49 in Zambia is 16.6% while 17% of women had sexual intercourse by age 15 and 69% by age 18 [4]. Statistics also showed that at the time of the 2018 Zambia Demographic and Health Survey (ZDHS), 29% of adolescents had begun childbearing [4]. With the low age at first sex among adolescents, contraceptive use is also low in this age group. In the period from 1996 to 2013/14, contraceptive use among adolescents increased by only 3.3% from 7.6% in 1996 to 10.9% in 2013/14 [5]. Data from the 2018 ZDHS also showed that only 12% of all women aged 15–19 years used modern contraceptives [4].

Contraceptives have been proven to help prevent unwanted pregnancies, early pregnancies, and unsafe abortions [6]. However, contraceptive use remains low, particularly in the least-developed countries, such as those in Africa [7,8,9]. Contraceptive use was as low as 40% and 33% in the least-developed countries and Africa, respectively [8]. Statistics revealed even lower figures among female adolescents [10,11], with the overall contraceptive prevalence rate (CPR) among female adolescents aged 15–19 in the developing world being about 21% for all methods (modern and traditional) [12].

This is despite the benefits of contraceptive use, such as the freedom to decide the number of children to have and how to space them, improvements in health-related outcomes, such as reduction in maternal mortality and infant mortality [13,14,15], and improvements in schooling and economic outcomes, especially for girls and women [16,17].

While there is a large body of evidence quantifying the magnitude of the problem surrounding contraceptive use among adolescent girls, the reasons for the low contraceptive use among adolescent girls are not well understood, especially in the Zambian context. These include motivators and influencers of adolescent girls’ contraceptive decisions. Various studies across Africa have shown that there are factors influencing adolescent girls’ decision to use contraceptives [6,18,19,20]. Among these are individual [6,21,22], parental [18,19,21], peer [20,21], partner [19,22], societal and community [6,18,20], and institutional and environmental [6,23] influences that contribute positively and negatively to adolescents’ decision to either use or not use contraceptives. In Zambia, however, there is limited information on what influences contraceptive decisions among adolescent girls aged 15–19 years. Therefore, this study sought to explore and understand what motivates and influences adolescent girls’ decision making regarding contraceptive use through qualitative methods.

## 2. Material and Methods

### 2.1. Study Design

This study was conducted through a qualitative exploratory design that sought to understand the motivators and influencers of adolescent girls’ decisions on whether or not to use contraceptives. Influencers in this study included people and institutions that encourage or discourage contraceptive use while motivators were reasons why adolescent girls decide to use or not use contraceptives.

### 2.2. Study Setting

The study was conducted in Chongwe and Lusaka districts in Lusaka province and Kasama and Luwingu districts in northern provinces. The two districts in each province were purposively selected based on whether they are predominantly rural or urban. Kasama and Lusaka districts in Northern and Lusaka provinces represented urban, while Chongwe (Lusaka province) and Luwingu (Northern province) represented rural.

### 2.3. Selection of Study Sites

In this study, one health facility with a functional and active Youth Friendly Services (YFS) corner was selected in each district with the help of the District Adolescent Health Coordinator as the site for the focus group discussion. These corners are operated by the Ministry of Health and are run through health facilities where space is provided to provide sexual and reproductive health IEC materials and products, such as condoms and contraceptives.

### 2.4. Study Participants and Recruitment

Adolescent girls who were recruited for the FGDs and KIIs were aged between 15 and 19 years. Purposive sampling, specifically maximum variation sampling, was employed in selecting participants for both FGDs and KIIs. Peer educators from the selected health facilities assisted with recruitment by administering the recruitment screening tool while the research assistants and principal investigators decided which participants were recruited based on the eligibility criteria. The criteria included only female participants who were residents of the study sites, aged between 15 and 19 years and accessed services from the youth-friendly spaces were included in the study. Maximum variation sampling ensured that there was diversity in the groups considering age, knowledge of contraceptives, and education levels.

### 2.5. Data Collection

In total, 7 focus group discussions (FGDs), with each having between 6 and 12 participants, and 3 key informant interviews (KIIs), were undertaken, each lasting between 60 and 90 min. KIIs were conducted with participants who were not comfortable with participating in the focus group discussions. All these involved adolescent girls aged between 15 and 19 years. The FGDs and KIIs were all conducted by two research assistants (RAs) trained and experienced in conducting qualitative research, who were supervised by the principal investigator. The RAs were experienced in qualitative research and were also trained and oriented in this study. All the research assistants had the required understanding of the local language and context of the study areas. The FGDs and KIIs were audio recorded, with participant permission, predominantly in English, with some participants opting for local languages—Bemba and Nyanja.

### 2.6. Data Analysis

Thematic analysis was used to analyse the data from both KIIs and FGDs. Thematic analysis is a method used to identify, analyse, and report patterns or themes observed within data [24]. It helps with organising and describing the data set in detail and interpreting different facets of the research topic. Components of the interviews that were recorded in the local language were transcribed and translated verbatim into English by the research assistants. NVivo version 12 pro (QSR International) was used to manage and organise the data, including data from both the KIIs and FGDs. Through an iterative process, one master code list was developed by the researchers. The process involved constantly reviewing the codes as well as the transcripts throughout the analysis to establish connections within the themes. Through this process, emerging themes were reviewed and, where needed, were merged with other similar themes until the main themes and subthemes were developed, which formed the master code list. Participant verbatim quotes were also used to ensure the credibility of the findings. The social–ecological model of health promotion [25] was used to categorise the themes. Health behaviour and promotion, according to their model, are interrelated and occur at multiple levels, including the individual, interpersonal, institutional, community, and policy levels [25]. Therefore, the results of this study are presented and discussed in line with the following themes, individual, partner, peer, parental, and societal influences, and how they influence adolescent girls’ contraceptive decisions.

## 3. Results

The study included a total of 70 adolescents across seven focus group discussions and three key informant interviews. Participants included a mix of adolescent girls who were sexually active and not sexually active, some who were married, and some who had children but were not married. There were also some who were in school and some who were not in school. Table 1, below, summarises the distribution of participants and the study site for each group and the geographical location.

Results from the seven focus group discussions and three key informant interviews revealed various motivators and influencers for both the decision to use and/or not use contraceptives among adolescent girls. Under the motivators for contraceptive use, there were six subthemes, namely, preventing pregnancy; fear of STIs, child spacing, and fear of having more children; fear of disturbing education; preventing worsening poverty and lack of family support; and peer pressure and influence from peers. Under the theme motivators for non-use of contraceptives were three subthemes and these were side effects of contraceptives; fear of infertility; being mocked by friends and peers. These themes and subthemes are detailed further below.

### 3.1. Motivators or Reasons for Contraceptive Use

#### 3.1.1. Fear of Becoming Pregnant and Fear of Disturbing Their Education

One of the key motivators for contraceptive use among adolescent girls was to prevent pregnancy. Adolescent girls were afraid of getting pregnant and, thus, resorted to using contraceptives, especially if they had observed their friends falling pregnant.


*“When you see your friend get pregnant or get sick at a young [age], you are motivated to use contraceptives so that you do not get pregnant.”*
[FGD 2, participant 5, Chikoyi Clinic, Luwingu]


*“To protect myself from getting pregnant, because if I do not get that injection, I can get pregnant quickly.” *
[FGD 2, participant 1, Chongwe Clinic, Chongwe]

The desire to prevent pregnancy was also linked to their fear of disrupting their education. For adolescents who were still in school, the fear of disrupting education was an added motivation for using contraceptives. Becoming pregnant would lead to them having to drop out of school for the duration of the pregnancy and also post-delivery so they opted to use contraceptives to prevent pregnancy and, thus, remain in school.


*“You just make a decision on what you want in life such that you decide whether you want to get pregnant there and then or first go to school and live better. So, I have decided to use contraceptives.” *
[FGD participant 3, Chongwe Clinic, Chongwe]


*“Some of us use contraceptives because say you are at school, I do not want to stop school because of pregnancy. I tend to use contraceptives while enjoying the sex till I finish school.” *
[FGD 1, participant 2, Location Urban Clinic, Kasama]

#### 3.1.2. Fear of Sexually Transmitted Diseases

The fear of contracting sexually transmitted diseases such as HIV/AIDS was also a strong motivator for using contraceptives among adolescent girls.


*“We feel they [condoms] are safe to prevent us from getting diseases such as STIs.” *
[FGD 1, participant 9, Location Urban Clinic, Kasama]


*“I think we just get scared of getting HIV and AIDS” *
[KII 3 participant, Mtendere Clinic, Lusaka]

Most of the adolescents expressed both the fear of becoming pregnant and the fear of diseases as the main motivators for their use of contraceptives.


*“To avoid diseases and pregnancies” *
[FGD 2, participant 3, Namalundu Clinic, Kasama]

#### 3.1.3. Child Spacing and Fear of Having More Children

Fear of having more children and child spacing were key motivators for contraceptive use, particularly among adolescents who had a child. Adolescents who already had a child started using contraceptives to prevent having more children. For some adolescents, fear of having a second child was what drove them to start using contraceptives.


*“For me, I decided to get it [injectable contraceptive] so that I should not fall pregnant immediately because I have a child.” *
[FGD 2, participant 1, Chongwe Clinic, Chongwe]


*“I looked at my child, she’s still young and so I’m scared to get pregnant again.” *
[KII 1, participant, Mtendere Clinic, Lusaka]

Among those who were married, child spacing was also one of the reasons why they chose to use contraceptives.


*“Not wanting to have children without spacing is what influenced me.” *
[FGD 2, participant 5, Chongwe Clinic, Chongwe]

#### 3.1.4. Prevent the Worsening of Poverty and Lack of Family Support

For some, the thought of the socio-economic situation at home or the possibility of struggling to take care of the child, or even the possibility of lack of support from family members if they fall pregnant, was motivation enough for them to use contraceptives. Having an extra mouth to feed in a household that is already struggling motivated adolescent girls to use contraceptives.


*“If I am not ready to become a parent, I might make my child suffer for nothing. That is the main reason.” *
[FGD1, participant 1, Chongwe Clinic, Chongwe]


*“Some [girls] it’s because of their social economic status at home. You will find that becoming pregnant is a cost they cannot afford.” *
[FGD 1, participant 4, Location Urban Clinic, Kasama]


*“To reduce poverty, because you find that at home, they are not managing to feed everyone, so if I bring a baby, it would be worse.” *
[FGD 2, participant 2, Chongwe Clinic, Chongwe]

Having no support from the family when they fall pregnant was also added motivation to use contraceptives for some of the adolescent girls.


*“People from home stop sponsoring [providing support] you once you fall pregnant.” *
[FGD 1, participant 9, Chongwe Clinic, Chongwe]

### 3.2. Motivations or Reasons for Non-Use of Contraceptives

Some adolescents expressed various reasons why they chose not to use contraceptives. These are presented below.

#### 3.2.1. Side Effects of Contraceptives

The side effects of the contraceptives that adolescent girls experienced were key motivations for the non-use of contraceptives, specifically discontinuation of use. Various side effects that they experienced were mentioned with prominent side effects, including bleeding, delayed or extended periods, headaches, numbness in the arm, and weight gain.


*“When I was using Depo, I used to gain [weight] and my stomach would bulge up, so that was why I decided to stop.” *
[FGD 1, participant 2, Chongwe Clinic, Chongwe]


*“The challenges I went through were unending periods, overweight too.” *
[FGD1, participant 2, Chongwe Clinic, Chongwe]


*“Periods never end when I get the injection for 2 months. Sometimes the hand which receives the injection freezes.” *
[FGD 1, participant 7, Chongwe Clinic, Chongwe]


*“When I first got the injection I had an unending flu, delayed with periods.” *
[FGD 1, participant 2, Chongwe Clinic, Chongwe]

#### 3.2.2. Fear of Infertility

What adolescents hear about contraceptives also gave them a reason not to use them. Some adolescents reported hearing that contraceptives would cause them not to conceive. They heard that using contraceptives such as the pill would cause them to become barren.


*“The things people say about contraceptives. For example, people say when you take a pill, you will become barren, so that discourages us from getting them” *
[FGD 2 Participant 4 Chikoyi Clinic, Luwingu]


*“People advise that getting contraceptives when you do not have a child means you will never conceive.” *
[FGD 1, Participant 9 Chongwe Clinic, Chongwe]

#### 3.2.3. Mocking by Friends and Peers

Adolescents reported that friends and peers also motivated them not to use contraceptives. Mocking by friends and peers also deterred some adolescent girls from using contraceptives.


*“Some of the youths look down on their friends who get contraceptives, and they usually mock them, so this makes most of them shun away.” *
[FGD 1, Participant 9 Chongwe Clinic, Chongwe]

### 3.3. Influencers for Use and Non-Use

Although discussions on actual contraceptive use are typically between girls and their boyfriends or partners, their use of contraceptives is influenced by other factors, both positively and negatively. These influencers shape what the girls know and how the girls view contraceptives. This influence can be direct or indirect and, thus, influences the girls’ decisions on whether they will use contraceptives. In this study, we found that influencers of contraceptive use among adolescents are intrapersonal, interpersonal, and organisational. At the intrapersonal level, these are related to the personal influences that shape their decisions while interpersonal influences include parents, friends and peers, family members, and partners or boyfriends. Organisational influencers include churches and religious groups.

### 3.4. Intrapersonal Influencers

*Personal:* For some adolescents, the motivation or influence to use contraceptives was a personal decision. The decision to use contraceptives is something that came from within themselves based on personal beliefs or experiences. One adolescent started using contraceptives because she cannot afford to take care of a child while another started using contraceptives for fear of labour and maternal mortality.


*“For me, it was myself that initiated the idea of using contraceptives. I told myself that at my age I cannot afford to take care of a child. So, I decided to start using contraceptives.” *
[FGD 1, Participant 6 Location Urban Clinic, Kasama]


*“For me, I fear labour as people die during labour. So, to avoid pregnancy I use a contraceptive.” *
[FGD 2, Participant 10 Chikoyi Clinic, Luwingu]

With regard to initiating the decision to use contraceptives with their sexual partners, there are instances where adolescent girls broached the discussion and made the decision to use contraceptives. This is primarily because they are the ones who are afraid of becoming pregnant and, sometimes, they use contraceptives without their partner knowing about it.


*“It is the girl [who starts the decision] as we are the ones who fear to get pregnant.” *
[FGD 2, Participant 3 Chikoyi Clinic, Luwingu]

Further, if they have unprotected sex, they feel that it is up to them to protect themselves by using emergency contraceptives.


*“It’s the lady. After the sex, you find the guy goes and it’s up to the lady to know how to protect herself from pregnancy.” *
[FGD 2, Participant 7 Namalundu Clinic, Kasama]

In some cases, both the adolescent girls and their boyfriends initiated the decision for contraceptive use together, after discussing it. They discussed together and planned the use of contraceptives, and the boyfriends also reminded the girls when they were due for their injections.


*“We both discuss [contraceptive use] then come up with a decision [to use contraceptives].” *
[FGD 2, Participant 1 Chongwe Clinic, Chongwe]


*“We exchange [initiating the discussion to use contraceptives], it can be me today and he will initiate it [the discussion] the next day. He used to remind me to go and get injections when I used to be on Depo.” *
[FGD 1, Participant 2 Chongwe Clinic, Chongwe]

### 3.5. Interpersonal Influencers

These mainly included people whom the adolescent girls interact with in their daily lives, including parents, friends and peers, boyfriends, and partners, as well as family members.

*Parents:* Parents were major influencers of contraceptive use among adolescent girls. Parents, though indirectly, have a positive influence on adolescents to use contraceptives through threats of repercussions should the girls fall pregnant. The fear of the consequences of falling pregnant based on the threats and warnings from their parents, particularly mothers, is a strong influence on them using contraceptives.


*“I had a friend whose mom and my mom were friends. That friend of mine got pregnant and my mom was told. Immediately she came to me and said your friend is pregnant, should you also bring a pregnancy here, you will see what I will do to you. This motivated me to take a contraceptive.” *
[FGD 1, Participant 2 Location Urban Clinic, Kasama]


*“For parents, they say if you get pregnant you will leave this place, so we take contraceptives in fear of our parents’ reactions.” *
[FGD 2, Participant 10 Chikoyi Clinic, Luwingu]


*“Yes, because the way they talk that if you get pregnant, you will be killed or beaten and chased. So, this influenced me.” *
[FGD 1, Participant 2 Chongwe Clinic, Chongwe]

Fathers also influence adolescents to use contraceptives through what they say to the girls.


*“The way my father talks that if I ever try to get pregnant, I will have a premature baby. When I hear this, I always get scared and end up getting contraceptives.” *
[FGD 1, participant 3 Chongwe Clinic, Chongwe]

*Friends and peers:* Friends and peers have a strong influence on contraceptive use among adolescent girls. Friends and peers have influenced contraceptive use either directly by being a source of encouragement or by reminding them to take the contraceptives. According to one girl,


*“It was my friend. After I had sex with my boyfriend, I went to my friend, told her what happened. Then she told me about morning after pill.” *
[FGD 1, Participant 6 Location Urban Clinic, Kasama]


*“We do influence each other by reminding each other to go and get the contraceptives.” *
[FGD 1, Participant 9 Chongwe Clinic, Chongwe]

Friends played a role in influencing other adolescents to use contraceptives. This influence came in two forms, namely: through hearing about contraceptives from their friends or observing their friends using contraceptives, even without full knowledge or understanding of different types of contraceptives and possible side effects thereof. Based on what they observed and also their friends’ experiences, they made decisions to also use contraceptives, often using the same methods that their friends were using.


*“I heard from people that it [using contraceptives] is nice, and I decided to try.” *
[FGD 1, participant 2, Chongwe Clinic, Chongwe]


*“For me, it was peer pressure. My friends influenced me to do it.” *
[FGD 2, participant 1, Chongwe Clinic, Chongwe]


*“It’s not everyone that takes contraceptives because they know what they are doing. Sometimes it’s because of peer pressure or influence from friends. When they see their friends get Jadelle, they also feel they can insert Jadelle. They do it without knowledge of why they are doing it.” *
[FGD 1, participant 7, Location Urban Clinic, Kasama]

The fear of discrimination from their friends and peers if they fell pregnant also influenced some adolescents to use contraceptives. Being shunned if they fell pregnant spurred them to use contraceptives. As one girl put it,


*“There is fear of discrimination from friends. They may not want to be found with you if you get pregnant.” *
[FGD 2, Participant 8 Namalundu Clinic, Luwingu]

Having a wedding is usually a source of pride, not just for young women but the family as well. Most adolescents who become pregnant before getting married usually do not have weddings. Therefore, for some girls, the possibility of getting married without a wedding if they fell pregnant was reason enough to use contraceptives.


*“Sometimes we also envy our friends who got married with weddings that is why we take contraceptives. We do not want to get pregnant as we fear we may not have weddings like our friends.” *
[FGD 1, Participant 7 Location Urban Clinic, Kasama]

*Family members****:*** Some family members also influence contraceptive use among adolescent girls. These include sisters, cousins, and aunts.


*“For me yes, because I share the same bed with her, so even if I get the condoms, she sees them and she uses them sometimes.” *
[FGD 1, Participant 3 Chongwe Clinic, Chongwe]


*“For me, my aunt is my friend. So, we talk about these things.” *
[FGD 1, Participant 3 Chongwe Clinic, Chongwe]

Even among adolescents who have a child, family members play a role in influencing them to use contraceptives.


*“My sister…. She told me to go to the clinic and get family planning so that I do not fall pregnant again.” *
[KII 1, Participant Mtendere Clinic, Lusaka]

*Boyfriends or Partners:* Sexual partners such as boyfriends also influence girls’ decisions to use contraceptives. They influence the decision to use contraceptives and the type of contraceptive used. Some girls were influenced by their boyfriends to use injectables, while others used condoms.


*“For me, my boyfriend who told me that he wanted us to be having sex on a regular basis. I agreed and told him that I did not want to get pregnant especially if we will be having sex regularly which was maybe once after a day except the days I am on my menstrual cycle. So, he asked for time to find me a contraceptive. He searched and found one. He explained to me what it was, and he took me to the clinic where I got it. The contraceptive was Jadelle.” *
[FGD 1 Participant 2 Location Urban Clinic, Kasama]


*“For me, my boyfriend told me he is not ready to get me pregnant, so he initiated the use of condoms.” *
[FGD 1 Participant 10 Location Urban Clinic, Kasama]


*“We remind each other but most of the times it is me who always makes sure that there is a condom because sometimes men pretend not to have condoms. It is up to him if he thinks that I am a prostitute, but I know it is my life and I need to take caution. I do not see anything wrong with a woman carrying condoms because you are protecting yourself at the end of the day.” *
[FGD 1 Participant 3 Chongwe Clinic, Chongwe]

Boyfriends and partners, when it came to using contraceptives, were the ones who initiated the decision to use contraceptives in some instances. They decided on contraceptive use as well as the type of contraceptive that was used and also enabled access to emergency contraceptives by giving the girls money to buy them.


*“It depends, if you plan to have sex with your boyfriend, he will be prepared with condoms. If it happens just at gunpoint, he will give you money to and buy a morning after pill. But if he does not, it’s up to you as a girl to think if you want to get pregnant or not. Then you will know what to do.” *
[FGD 1, Participant 7 Location Urban Clinic, Kasama]


*“Sometimes the guy will after the sex give a lady money to go and buy a morning after pill.” *
[FGD 2, Participant 3 Namalundu Clinic, Kasama]

### 3.6. Organisational Influencers

*Churches and religious groups:* Churches and religious groups also, in some way, influence contraceptive use among adolescents. The consequences of falling pregnant such as ex-communication from church positively influence adolescent girls’ decision to use contraceptives.


*“At my church, if you get pregnant, they make you stand in front of the church, and they ban you from going to church. So that makes us use contraceptives in fear of being banned from going to church.” *
[FGD 2, Participant 5 Chikoyi Clinic, Luwingu]


*“These days in churches they disfellowship young ladies who fall pregnant. So, to avoid that, they end up using contraceptives.” *
[FGD 2, Participant 8 Namalundu Clinic, Luwingu]


*“At our church, if you get pregnant, they stop you from going to church and taking part in church-related activities. So, we take contraceptives so that we do not get pregnant and be stopped from going to church.” *
[FGD 1, Participant 10 Chikoyi Clinic, Luwingu]

## 4. Discussion

Overall, the results from this study have shown that the decision-making process by adolescent girls on whether or not to use contraceptives is influenced by a multitude of factors, which can be best understood through the socio-ecological model (SEM). The model offers a basis to understand the dynamic relationship that exists between an individual and their environment as a determining factor of health behaviour. An individual’s health behaviour is not simply the result of the individual’s decision, but it is moulded by a multifaceted interaction of various internal and external influences in their environment, which may include (1) individual factors, (2) interpersonal relationships, (3) organizational entities, (4) community factors, and (5) systems and policy [26]. Results from this study demonstrated that there are various motivators for the use and non-use of contraceptives among adolescent girls and the contraceptive decisions they make are also influenced by various intrapersonal, interpersonal, and societal factors. Most of the motivators for the use and non-use of contraceptives are intrapersonal while most of the influencers are intrapersonal and organisational, which raises the question whether adolescent girls actually have the power to make autonomous contraceptive decisions. These are discussed further.

### 4.1. Motivators for Contraceptive Use

Intrapersonal motivators: Prevention of pregnancy was a major motivation for using contraceptives and other studies have reported similar findings [18,27,28]. It can be linked to other motivations, such as the fear of the consequences of teenage pregnancy. They want to avoid socio-economic consequences, such as disturbing their education, as they will inevitably drop out of school, lack of family support and abandonment, alienation, and social as well as internalised stigma, and this is consistent with other studies [19,22,28,29,30]. Adolescents have the desire to prioritise their education as well as not bring disappointment and shame to their families and endure the social stigma that comes with being teenage mothers [28,29]. Poverty, or the fear of worsening it, is a strong motivator for adolescent girls to use contraceptives.

Fear of diseases is also another motivation for contraceptive use, particularly condoms. Adolescent girls who did not want to use hormonal contraceptives opted to use condoms, particularly male condoms. Comparable findings have been reported in other studies [27,31,32,33]. Condoms are viewed as both a means of protection against diseases and prevention of pregnancy [33], although most use them for disease protection rather than the dual role that condoms provide (disease prevention and prevention of pregnancy) [34]. Our study did not investigate the non-mentioning of female condoms as a contraceptive of choice by adolescent girls.

Adolescents who had at least one child reported using contraceptives out of fear of having more children, and among married adolescents, as a way of spacing their children, and this has been reported in other studies [35]. Adolescent mothers are motivated to use contraceptives as they seek to avoid the socio-economic and possible health consequences of having more children, or children born close together. Poverty and failure to support more children provide added incentives for adolescent mothers to use contraceptives [36].

### 4.2. Motivators for Non-Use of Contraceptives [1,27,36,37,38]

The main reasons given by the adolescent girls for non-use of contraceptives centred around three main themes, which were fear of side effects, fear of infertility, and fear of mocking or stigma by their friends. These can be categorised as intrapersonal and interpersonal motivators.

Intrapersonal motivators for non-use: The adolescents mentioned various side effects that they have experienced from using contraceptives. Similar side effects have been reported in other studies, including menstrual cycle irregularities [6,20,23], sickness, including abdominal pain [18,19,23], changes in weight (either extreme weight gain or loss) [6,19,21,22,23], menstrual cycle and colour of blood, spotting, constant bleeding [21,23], amongst others. Other issues mentioned include body weakness, dizziness, fainting, blood clots during menstruation, miscarriages, becoming epileptic [6], heavy bleeding [6,18], cancer [18], reduced libido [19,23], and paralysis [21]. This fear stems from what those who have used contraceptives have experienced and also what those who have not used contraceptives have heard from those who have used them. These fears remain a strong motivation for the non-use of contraceptives, especially since the experience differs from individual to individual. Some may experience severe adverse reactions while others may not, and as these experiences are shared, and coupled with misconceptions about contraceptives, this fear of experiencing adverse side effects remains prevalent among not just adolescents, but women in general.

The fear of contraceptives causing infertility was a major reason for non-use [35,39,40,41]. The belief that modern contraceptive use causes infertility has been documented in other studies in sub-Saharan Africa [35,39,40,41,42,43,44,45,46]. In most societies in Africa, women face a lot of pressure to have children. For some, having children and motherhood are viewed positively and thought to give the woman respect and an elevated social status in the community [47]. Adolescent girls and women who are unable to bear children may face challenges socially and culturally, including disruption of their relationships or marriages, boyfriends or husbands resorting to polygamy, divorce, or promiscuity by the boyfriend or husband [41,48]. Therefore, when their fertility is threatened or is seemingly affected through contraceptive use, these adolescent girls, and women in general, will opt to not use contraceptives or discontinue use where they may have started already in favour of those perceived not to affect fertility such as condoms.

Interpersonal motivators for non-use: Stigma associated with contraceptive use among adolescents has been reported in other studies [27,30]. This stems from the association of contraceptive use with promiscuousness. Where contraceptive use among adolescents becomes common knowledge, they may be ridiculed and labelled ‘bad girls’ who have loose morals, as reported in similar studies [27,30,48,49]. Such mocking and judgement from their peers may lead to exclusion from their social circles and them being at the centre of gossip among their peers, including those who are married [6,20]. Therefore, to avoid being the centre of ridicule, gossip, and victims of social exclusion and for them to maintain their social circles and social life, adolescents may lean towards the non-use of contraceptives, that is, if they do not opt to hide their usage of contraceptives.

### 4.3. Influencers for Use/Non-Use [20,35,50]

Various factors influence adolescents’ decision whether or not to use contraceptives and the main influencers included their parents, peers and friends, family members, partners (interpersonal influencers), as well as churches and religious groups (organisational influencers).

Interpersonal influencers: Parents, particularly mothers, play a key role in influencing the contraceptive decisions of adolescent girls, as reported in several other studies [6,19,20,21,23], in most cases indirectly [21]. Positive parental influences involve conversations, mostly with mothers, regarding contraceptives, which help them choose, particularly if they discuss contraceptives with their mothers before choosing a method [6]. Negative influences stem from negative attitudes that parents have towards contraceptive use, which are tied to the disapproval of adolescents engaging in sexual activities. Some parents do not discuss contraceptive use due primarily to their cultural and religious beliefs as well as the belief that they are meant for adults [18]. Adolescent girls are afraid that their parents would perceive them as bad girls for engaging in sex, which, in their parents’ view, is sinful behaviour and disrespectful to the parents [6]. Repercussions, such as parental disappointment and being driven from home or disowned, should parents discover their contraceptive use tend to incline adolescent girls not to use contraceptives [6,18]. Beyond just influencing the decision to use contraceptives, parents, particularly mothers, also influence the choice of contraceptive methods, usually advising against hormonal methods that are thought to affect fertility [19].

Sexual partners can both encourage and discourage contraceptive use. Partners tend to encourage contraceptive use to avoid the embarrassment of making a young girl pregnant [19], while others do it to engage in unprotected sex while avoiding pregnancy. Where partners encourage contraceptives, they play a key role in deciding the contraceptive method of choice [20]. Partners, however, also discourage contraceptive use through coercive tactics to ensure the non-use, misuse, or discontinuation of contraceptive use [47]. They discourage adolescent girls from using contraceptives due to concerns about contraceptives and their perceived effect on future fertility [19]. Adolescents also avoid using contraceptives for fear of their partners thinking they have multiple sexual partners [20]. Even though adolescent girls discuss contraceptive use with their intimate partners, these conversations are usually to provide approval for contraceptive use [20,51], which is linked to the unequal gender power dynamics in relationships, with males ultimately controlling the decision making on contraceptive use [51]. They also influence the choice of contraceptive method, which typically aligns with their preferred method [52].

Friends and peers have a major influence on adolescent girls’ decisions regarding contraceptive use, both positively and negatively [27,36,37]. They are a key channel through which information on contraceptives is shared [1]. They, through their experience, influence adolescent girls’ contraceptive decisions. As confidants, friends and peers are usually the first to know about intentions to have sex. Therefore, based on their shared fears of pregnancy, disruption of education and other such fears, coupled with some experience or knowledge about contraceptives that the friends may have, the friends have a major influence on the decision to use contraceptives. Therefore, adolescents who see their friends as contraceptive users are more likely to use contraceptives themselves [38]. As positive influences, they provide encouragement, share experiences, and also offer reminders when contraceptives are due [27,36,37]. Adolescent girls tend to use contraceptives because their friends are using them [22], and they also share contraceptives, such as condoms, with each other. As negative influences, friends and peers associate contraceptive use with promiscuous behaviour, which can discourage contraceptive use [20]. Friends may also distance themselves from those discovered to use contraceptives, so to avoid being excluded from social groups, adolescent girls may not use or may discontinue the use of contraceptives [6,27,30].

Family members, other than parents, also influence the contraceptive decisions of adolescent girls [27]. These typically include sisters, cousins, aunts, and grandmothers, depending on whether the adolescent girls were married or not. Older women encouraged contraceptive use, recommending different methods, depending on whether the girl is married or not [53]. Adolescents whose family members were more positive and had a liberal attitude and encouraged contraceptive use found it easier to use contraceptives [54]. However, family members can also discourage the use of contraceptives through the disapproval of their use [27], especially amidst perceived effects on future fertility [41].

Organisational influencers: Churches and religious groups also influence adolescent girls’ decisions to use contraceptives, in some cases, a negative influence, through their religious teachings, which are against sex, particularly among young and unmarried women, with those discovered to be sexually active being shunned or rejected by their religious communities for sex and its consequences [47]. Religious groups tend to discourage the use of contraceptives, particularly among adolescents, considering it a sinful act against God and viewed in the same light as abortion [6]. However, other studies show that there are others who permit contraceptive use with regard to child spacing that positively influence contraceptive use [55]. These results and others from other studies show that religion does play a role in influencing contraceptive decisions, whether negative or positive, and should be leveraged to improve the uptake of contraceptives.

### 4.4. Adolescent Girls’ Contraceptive Decision-Making Power

This study has shown that adolescent girls’ decision making regarding contraceptive use is the result of a combination of an intimate, close, and long-term relationship; positive self-esteem; an internal locus of control; and favourable family, partner, and peer influences, and when these characteristics are conducive, decision making takes place, with contraceptive use being the final outcome [56]. However, influencers of contraceptive decisions also affect the power that these girls have to make these contraceptive decisions. There exist inherently unequal power dynamics in relationships with their sexual partners. These may result from the age difference, financial dependence of the girls on their partners, etc., and usually take away the autonomous decision-making power that the girls may have due to the potential repercussions should they go against what their partner wants. Friends and family also limit the decision-making power that adolescent girls may have regarding contraceptive use. Through their influence, adolescent girls may feel like they have no option but to go with what their friends and family are doing or saying, thus relinquishing their decision-making power.

#### Study Limitations

The study’s findings may have been influenced by social desirability bias, and the sensitive nature of the topic may have hindered information sharing by the participants. Additionally, the study sought to capture broad and relevant themes regarding adolescent girls’ contraceptive decisions to teenage experiences rather than draw conclusions from the FGDs and KIIs based on a representative sample. However, trained and experienced researchers conducted the interviews and discussions to mitigate bias. The study team also attempted to include more varied perspectives in the focus groups by implementing a more inclusive recruiting method and ensuring that all participants’ opinions were heard.

The recruitment of participants from the youth-friendly services/corners may have biased recruitment to those who are inclined to seek health services from health facilities or live in proximity to the location of the centres may affect the application of these findings to dissimilar locations from where this study was conducted. However, these findings are important for understanding the motivations and influences of adolescent girls’ contraceptive decisions. Future studies should consider a different approach in the recruitment to strengthen the generalizability of the findings.

## 5. Conclusions

Adolescents are motivated to use contraceptives by their fear of pregnancy, fear of diseases, and also socio-economic consequences that come with teenage pregnancy, such as dropping out of school, social stigma, and associated poverty. Their decision to use contraceptives is also influenced by different factors, including their parents, peers and friends, family members, partners, as well as churches and religious groups. This brings into question whether adolescent girls have the power to make decisions regarding contraceptive use, as the decision to use contraceptives as well as the choice of method are influenced by persons other than themselves. There is a need to understand and account for all these motivations and influences in the development of intervention programmes targeting increasing contraceptive use in this age group. It is vital to take an all-inclusive approach in developing interventions by including various influencers, including at institutional and policy levels, even as we seek to empower adolescents and give them autonomy to make contraceptive decisions. Parents, family members, and community and religious groups need to be engaged and sensitized on the importance and benefits of contraceptive use and dispel misinformation about contraceptives. Providing the correct information to counter misinformation is essential to improving contraceptive use for first-time users and continued use for those already using contraceptives.

## Figures and Tables

**Table 1 ijerph-20-03614-t001:** Characteristics of the participants.

Type of Interview	Facility Name	Total Number of Participants	Age Group	District	Geographical Location
FGD	Chikoyi Clinic 1	10	15–19	Luwingu	Rural
FGD	Chikoyi Clinic 2	10	15–19	Luwingu	Rural
FGD	Chongwe Clinic 1	9	15–19	Chongwe	Rural
FGD	Chongwe Clinic 2	6	15–19	Chongwe	Rural
FGD	Location Urban Clinic	10	15–19	Kasama	Urban
FGD	Namalundu Clinic	10	15–19	Kasama	Urban
FGD	Mtendere Clinic	12	15–19	Lusaka	Urban
KII	Mtendere Clinic	1	15–19	Lusaka	Urban
KII	Mtendere Clinic	1	15–19	Lusaka	Urban
KII	Mtendere Clinic	1	15–19	Lusaka	Urban
**Total**		**70**			

FGD: 7, KII: 3.

## Data Availability

Data can be accessed upon request through the corresponding author.

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
