# Peer review of "Motivators and Influencers of Adolescent Girls’ Decision Making Regarding Contraceptive Use in Four Districts of Zambia"

_ijerph, 2023, doi:10.3390/ijerph20043614_

Round 1

Reviewer 1 Report

General Comment

The proposed manuscript offers interesting and much needed qualitative research insights on determinants of contraceptive (non) use among adolescent girls living in Sub-Saharan Africa (specifically Zambia). While de qualitative approach is important to contextualize and interpret contraceptive patterns in this key population, and the paper is overall correctly organized, we believe some major revisions are needed to improve its scientific value.

As a general point, it seems the authors did not fully exploit the thickness of their qualitative data. The illustrative quotes selected tend to be short and reflect points that could be made out of quantitative surveys (e.g. “wanted to avoid getting pregnant”, “fear of side effects”) We suggest the authors select and analyze in depth longer excerpts from the FGD transcripts in order to provide added value to the existing body of evidence. (This is sketched out in the analysis of the “fear of poverty” findings but could be developed throughout the manuscript)

The theoretical model (SEM) for the analysis should be introduced in the Background and Methods section (it only appears in the Discussion) to clarify the position of motivators and influencers in contraceptive seeking behaviors.

Those two key categories should also be defined, and their relationship clarified in the analysis.

The authors need to address possible bias in participant recruitment and correlative threats to the generalizability of their findings: since they recruited participants among young women who frequented youth corners, is there a chance that the sample was based on adolescents with better access / who were more positively inclined towards SRH than the general population?

Were there any concerns that the “maximum variation sampling” could skew the power dynamics during the FGD (with older, more knowledgeable participants controlling the conversation or influencing the other participants)?

The Methods section does not address the Key Informants (KII) at all. Who were they? How were they recruited? Why were they added to the data collection process? How were there responses collected and analyzed?

A brief description of key demographic characteristics of the participants might be useful to highlight possible bias in the responses. In particular, how many participants were married / already had children? This is especially important because “fear of having more children” is listed as a key motivator.

Over the results section tends to repeat the same information multiple times in the same subsection. For example “fear of contracting the disease…The adolescents were afraid of getting sexually transmitted diseases… Most adolescents expressed the fear of getting pregnant and the fear of disease as their main motivators” appear in one short paragraph.

l. 137  What is the difference between “peer pressure” and “influence from peers”?

Also, influence from peers seems to work both towards use and non-use of contraception? Shouldn’t peers be considered “influencers” rather than “motivators”?

This again underlines the importance of clearly defining the two categories in the Background and Methods section, and to clarify their relationship through a clear theoretical behavior framework.

l. 124  Mentions “mocking by friends and peers” what was the reasons given for “mocking”? Assumed promiscuity? Other?

l. 247  “Initiator of contraception”  This concept is introduced mid- “Results” section and it’s unclear how it relates to either “motivators” or “influencers”

In addition, the findings are extremely vague (“it can be A, or B, or A and B “depending on the situation”) and do not contribute to the evidence-basis.

More generally, we recommend avoiding the “laundry list” effect (“some said… some said…”) of listing findings without organizing them to highlight majority and minority opinions and unexpected findings.

Regarding the influencers in particular, it would be useful to clarify which ones were overall positive support or potential barriers towards contraceptive use. It is interesting that in this group, parents and older family members seem supporting although it would be interesting to further distinguish between “not wanting their adolescent girl to become pregnant” and “being actively supportive of contraceptive use”

Did the FGD ask follow-up questions regarding who among the influencers had the strongest influence on the participants? It might be worth clarifying who adolescent girls hear an opinion from and who they actually listen to?

Overall, the reviewer would like to point out once again that the relationship between influencers and motivators are not clearly established in the analysis. The latter in particular are not coming out of nowhere: who circulates the information / motivation used by the participants to use contraception or not? Who shapes their knowledge around SRH?

What was the operational difference between “friends” and “peers” in the analysis? Did they function differently as influencers?

The Discussion tends to repeat findings from the previous section and introduces points from other studies that were not addressed in the research. It would be worth rewriting this section to engage more clearly with the existing literature and the contribution of this specific research to the existing body of evidence.

In particular, the authors introduce a very interesting point about teenagers’ “real” empowerment to make any contraceptive decision and it would be a welcome question to develop more in the entire Discussion section.

In the conclusion (l. 575-576) the authors indicate that their manuscript “provided strategies and scientific information on the implications of contraceptive usage, and to better inform policy development and strengthen the promotion of youth-friendly service provision”, but only some hints of strategic orientations are being built out of the evidence presented. The few recommendations articulated in the paper should be further detailed and their relation to the research findings highlighted more clearly.

Finally, the manuscript does not include a “Limitation” section or any mention of limits to the validity of the research design and findings. We strongly suggest the authors to develop such a paragraph.

In addition to these cross-cutting issues, a number of grammatical and syntax errors could be corrected to improve the manuscript. We have listed some below but recommend the use of a professional academic editor to ensure the quality

Abstract

l. 11    « Low contraceptive in Sub-Saharan…”: there seem to be a word missing.

Introduction

l. 64    “This is inclusive of…”: sentence is a bit awkward, try and reformulate?      

l. 67    “peers

Materials and Methods

l. 79    We suggest tightening up this paragraph to avoid repetitions of “rural” and “urban

l. 87    How were “active” and “functional” defined for Youth Friendly Services corner? Did the researchers review service statistics? Observe frequentation? Relied on local health authorities’ recommendations?

l.139 – 142    This sentence is extremely unclear, please review structure. 

Discussion

l. 466-468      Almost the same sentence repeated twice in a row.

Author Response

RESPONSES TO REVIEWER 1 COMMENTS

Reviewer Comment 1: The proposed manuscript offers interesting and much needed qualitative research insights on determinants of contraceptive (non) use among adolescent girls living in Sub-Saharan Africa (specifically Zambia). While de qualitative approach is important to contextualize and interpret contraceptive patterns in this key population, and the paper is overall correctly organized, we believe some major revisions are needed to improve its scientific value.   

Response: Thank you for taking the time to review our manuscript and contributing to improving it.

Reviewer Comment 2: As a general point, it seems the authors did not fully exploit the thickness of their qualitative data. The illustrative quotes selected tend to be short and reflect points that could be made out of quantitative surveys (e.g. “wanted to avoid getting pregnant”, “fear of side effects”) We suggest the authors select and analyze in depth longer excerpts from the FGD transcripts in order to provide added value to the existing body of evidence. (This is sketched out in the analysis of the “fear of poverty” findings but could be developed throughout the manuscript)

Response: We have restructured sections of the results to ensure that we explore the thickness of the results and also reduce of the repetition of similar subthemes.

Reviewer Comment 3: The theoretical model (SEM) for the analysis should be introduced in the Background and Methods section (it only appears in the Discussion) to clarify the position of motivators and influencers in contraceptive seeking behaviors.                                                     

Response: The SEM has been added in the methods section to help clarify the position of motivators and influencers in contraceptive-seeking behaviours.

Reviewer Comment 4: Those two key categories should also be defined, and their relationship clarified in the analysis.        

Response: It is not clear which two categories are being referred to by the reviewer.

Reviewer Comment 5: The authors need to address possible bias in participant recruitment and correlative threats to the generalizability of their findings: since they recruited participants among young women who frequented youth corners, is there a chance that the sample was based on adolescents with better access / who were more positively inclined towards SRH than the general population?                  

Response: We have added a limitations section that addresses potential bias resulting from how the recruitment was done and the resulting implication on the generalisability of the findings.

Reviewer Comment 6: Were there any concerns that the “maximum variation sampling” could skew the power dynamics during the FGD (with older, more knowledgeable participants controlling the conversation or influencing the other participants)?

Response: The issue of power dynamics was a point of concern during the planning and recruitment of participants for the study. Therefore, we recruited and trained experienced research assistants who can identify potential power dynamics in the groups and mitigate this by giving all participants, including those who may seem timid, a voice in the discussion by soliciting their thoughts

Reviewer Comment 7:  The Methods section does not address the Key Informants (KII) at all. Who were they? How were they recruited? Why were they added to the data collection process? How were there responses collected and analyzed?                                                                  

Response: Descriptions and references to the KIIs have been added to the different sections in the methods to clarify why they were conducted, and how the recruitment, data collection and analysis were done.

Reviewer Comment 8:  A brief description of key demographic characteristics of the participants might be useful to highlight possible bias in the responses. In particular, how many participants were married / already had children? This is especially important because “fear of having more children” is listed as a key motivator.  

Response: During the recruitment phase, we missed the opportunity to collect information on marital status. The screening tool focused on whether they were in a sexual relationship, knowledge and use of contraceptives and other information. Information on having children arose from the discussions and was common across all FGDs in all locations.

Reviewer Comment 9: Over the results section tends to repeat the same information multiple times in the same subsection. For example, “fear of contracting the disease…The adolescents were afraid of getting sexually transmitted diseases… Most adolescents expressed the fear of getting pregnant and the fear of disease as their main motivators” appear in one short paragraph.                                   

Response: Section 3.1.2 has been revised to remove the repetition.

Reviewer Comment 10: l.137 What is the difference between “peer pressure” and “influence from peers”? Also, influence from peers seems to work both towards use and non-use of contraception? Shouldn’t peers be considered “influencers” rather than “motivators”? This again underlines the importance of clearly defining the two categories in the Background and Methods section, and to clarify their relationship through a clear theoretical behavior framework. l. 124  Mentions “mocking by friends and peers” what was the reasons given for “mocking”? Assumed promiscuity? Other?

Response: These are more or less the same. We have revised this and retained “Peer pressure”. We agree with this. Friends are more influencers than motivators of contraceptive use. We have thus removed section 3.1.5 and merged it with the “Friends and peers” subsection under section 3.6. Definitions of what influencers and motivators were in the context of this study gave been included under section 2.1 under the Methods and Materials section.

Reviewer Comment 11: l. 247  “Initiator of contraception” ß This concept is introduced mid- “Results” section and it’s unclear how it relates to either “motivators” or “influencers”. In addition, the findings are extremely vague (“it can be A, or B, or A and B “depending on the situation”) and do not contribute to the evidence-basis.   This falls under influencers of contraceptive use. Therefore, this section has been integrated into sections 3.4 and 3.5 under personal and boyfriend/partner respectively.

Response: By restructuring and merging with these sections, this has become clearer.

Reviewer Comment 12: More generally, we recommend avoiding the “laundry list” effect (“some said… some said…”) of listing findings without organizing them to highlight majority and minority opinions and unexpected findings.      

Response: The side effects listed were the most common ones mentioned by the participants and were consistent with what other studies have found.

Reviewer Comment 13: Regarding the influencers in particular, it would be useful to clarify which ones were overall positive support or potential barriers towards contraceptive use. It is interesting that in this group, parents and older family members seem supporting although it would be interesting to further distinguish between “not wanting their adolescent girl to become pregnant” and “being actively supportive of contraceptive use”      

Response: Older family members encouraged contraceptive use for adolescent girls who had a child already as a means of preventing pregnancy too soon after the first child/child spacing but discouraged use for those who had no children because of the fear of the effect of contraceptives on future fertility. Parents on the other hand, based on the threats made to the girls on the consequences should they fall pregnant could be seen as them not wanting their children to fall pregnant. Thus their influence is more indirect.

Reviewer Comment 13: Did the FGD ask follow-up questions regarding who among the influencers had the strongest influence on the participants? It might be worth clarifying who adolescent girls hear an opinion from and who they actually listen to?                                                                   

Response: We did not really explore this during the FGDs which may have further enriched our findings. We will endeavour to explore this further in future research on this topic.

Reviewer Comment 14: Overall, the reviewer would like to point out once again that the relationship between influencers and motivators are not clearly established in the analysis. The latter in particular are not coming out of nowhere: who circulates the information / motivation used by the participants to use contraception or not? Who shapes their knowledge around SRH?                                                             

Response: We have explained what motivators and influencers are and we believe this clarifies this as well. Motivators are reasons that adolescent girls have for using contraceptives while influencers are people in their lives who shape their contraceptive decisions.

Reviewer Comment 15: What was the operational difference between “friends” and “peers” in the analysis? Did they function differently as influencers?                              

Response: Friends would include those whom they spend time with while peers may include their agemates who may not necessarily be close friends.

Reviewer Comment 16: The Discussion tends to repeat findings from the previous section and introduces points from other studies that were not addressed in the research. It would be worth rewriting this section to engage more clearly with the existing literature and the contribution of this specific research to the existing body of evidence. In particular, the authors introduce a very interesting point about teenagers’ “real” empowerment to make any contraceptive decision and it would be a welcome question to develop more in the entire Discussion section.

Response: We have restructured the discussion to align with the structure of the results section. We have added a paragraph on decision making power and how that is affected by the various influencers on adolescent girls’ contraceptive decision making.

Reviewer Comment 17: In the conclusion (l. 575-576) the authors indicate that their manuscript “provided strategies and scientific information on the implications of contraceptive usage, and to better inform policy development and strengthen the promotion of youth-friendly service provision”, but only some hints of strategic orientations are being built out of the evidence presented. The few recommendations articulated in the paper should be further detailed and their relation to the research findings highlighted more clearly.                  

Response: We have made revisions to the conclusion and expanded on the recommendations that have been proposed.

Reviewer Comment 18: Finally, the manuscript does not include a “Limitation” section or any mention of limits to the validity of the research design and findings. We strongly suggest the authors to develop such a paragraph.

Response: We have added a section limitations has been added to the manuscript.

Reviewer Comment 19: In addition to these cross-cutting issues, a number of grammatical and syntax errors could be corrected to improve the manuscript. We have listed some below but recommend the use of a professional academic editor to ensure the quality                                         

Response: Grammatical and syntax errors have been addressed.

Abstract                                                                      

Reviewer Comment 20: l.11 « Low contraceptive in Sub-Saharan…”: there seem to be a word missing.  

Response: The missing word has been added.

Introduction                                                                

Reviewer Comment 21: l.64 “This is inclusive of…”: sentence is a bit awkward, try and reformulate?  

Response: This sentence has been reformulated to improve clarity.

Reviewer Comment 22: l. 67 “peers”                          

Response: This has been retained as peers since this is referring to peer influences on adolescents’ decisions regarding contraceptive use.

Materials and Methods                                               

Reviewer Comment 23: l. 79 We suggest tightening up this paragraph to avoid repetitions of “rural” and “urban   

Response: We have made some revisions based on the suggestion by the reviewers.

Reviewer Comment 24: l. 87 How were “active” and “functional” defined for Youth Friendly Services corner? Did the researchers review service statistics? Observe frequentation? Relied on local health authorities’ recommendations?        

Response: Determination of functional Youth Friendly Services corner was done based on the recommendation of the local health authorities. From their explanation, their determination was based on health facilities that had YFS corners that has adolescent peer educators who provided services such as condom distribution, and provision of IEC materials on HIV, STI, and contraceptives.

Reviewer Comment 25: l.139 – 142 This sentence is extremely unclear, please review structure. 

Response: Following the revision of the results section, this sentence has been removed.

Discussion                                                                 

Reviewer Comment 26: l. 466-468 Almost the same sentence repeated twice in a row.  

Response: One of the sentences has been deleted to remove the repetition.

Reviewer 2 Report

This article is very well done. The article was stimulating, and I found the topic very interesting. You successfully showed how a variety of interpersonal, medical, economic, social, and cultural considerations all influence teen contraceptive uptake.

Since the article is already very strong, I could use my time as a reviewer to really focus on small details to strengthen the paper even more, since there are no large problems to address.

Firstly, there are a few small typos; e.g. on line 11 it reads “Abstract: Low contraceptive …” but I think it should read “Abstract: Low contraceptive use in…”.

I would like more descriptives to set up your finding. I’m not sure where you are with your word-limit, but I would like to know if all of the girls who you interviewed sexually active? This information is vital.

Similarly, please add more general descriptives for context for the reader: What percentage of female teenagers are sexually active/have ever had sex (I believe this data is available in the 2018 DHS report). Similarly, some information about the average age at first marriage, or age at first childbirth would be helpful to give the reader more context.

You say you recruited participants from Youth Friendly Services. What is this organization? How do people come into contact with it? Please add a note in your discussion about the limited generalizability of your findings.

In your discussion, you had a section on “organizational influencers” where you discussed the role of churches. In the discussion section of the paper, you discussed how churches lead to lower contraceptive uptake by teaching against contraception. Churches only came up in your results, however, as an influencer that results in higher contraceptive uptake (to avoid the social shame of pregnancy). Please reconcile these two and make the results section consistent with your discussion.

Overall, very well done. I look forward to seeing this article in print.

Author Response

RESPONSES TO REVIEWER 2 COMMENTS

Reviewer Comment 1: This article is very well done. The article was stimulating, and I found the topic very interesting. You successfully showed how a variety of interpersonal, medical, economic, social, and cultural considerations all influence teen contraceptive uptake.                            

Response: Thank you very much for taking the time to review our work and contributing to improving our manuscript. We appreciate it and we’re happy you found it interesting and stimulating.

Reviewer Comment 2: Since the article is already very strong, I could use my time as a reviewer to really focus on small details to strengthen the paper even more, since there are no large problems to address.

Response: Thank you.

Reviewer Comment 3: Firstly, there are a few small typos; e.g. on line 11 it reads “Abstract: Low contraceptive …” but I think it should read “Abstract: Low contraceptive use in…”.         

Response: The typos have been addressed.

Reviewer Comment 4: I would like more descriptives to set up your finding. I’m not sure where you are with your word-limit, but I would like to know if all of the girls who you interviewed sexually active? This information is vital.          

Response: We have added some description on this in the results section (line 135-137)

Reviewer Comment 5: Similarly, please add more general descriptives for context for the reader: What percentage of female teenagers are sexually active/have ever had sex (I believe this data is available in the 2018 DHS report). Similarly, some information about the average age at first marriage, or age at first childbirth would be helpful to give the reader more context.                                                             

Response: Descriptives have been added in the introduction to provide some context

Reviewer Comment 6: You say you recruited participants from Youth Friendly Services. What is this organization? How do people come into contact with it? Please add a note in your discussion about the limited generalizability of your findings.             

Response: A sentence has been added to the study setting (section 2.3) describing the youth friendly service corners. We have also added a limitation section which also refers to generalisability of the findings.

Reviewer Comment 7: In your discussion, you had a section on “organizational influencers” where you discussed the role of churches. In the discussion section of the paper, you discussed how churches lead to lower contraceptive uptake by teaching against contraception. Churches only came up in your results, however, as an influencer that results in higher contraceptive uptake (to avoid the social shame of pregnancy). Please reconcile these two and make the results section consistent with your discussion.                       

Response: This has been harmonised. The positive influence raised in the discussion has been discussed in the context of findings from other studies.

Reviewer Comment 8: Overall, very well done. I look forward to seeing this article in print.         

Response: Thank you once again.

Round 2

Reviewer 1 Report

Thank you for this much improved version of your manuscript. We appreciate your thoughtfulness in answering our initial suggestions. 

Overall, the Discussion section still needs to be edited as it too long and repeats many of the findings presented under results.

The Limitations section needs to be expanded as the authors simply copied and pasted (verbatim) one of the suggestions made by the reviewers but the study has additional limitations in terms of validity and generalizability.

Throughout the revised manuscript, the authors have made a diligent effort to clarify “influencers” vs “motivators”, but now need to be more rigorous in looking at “intrapersonal” vs “interpersonal”.

We have also noted a smattering of grammar and syntax errors or confusing statements that we think should be addressed before this manuscript can be published. We are listing a sample of them below but we suggest that the Editorial Board give a thorough review and proofing of the manuscript before its publication.

Abstract

l. 12                Review sentence construction as it currently reads as if “low contraceptive prevalence prevents […] unsafe abortion”

Main manuscript

Review first sentence. It appears too broad and it’s unclear that adolescents globally bear a heavier burden of disease (than say, children under 5). Perhaps refocus on adolescent health issues?

l. 58    Review “low sexual debut”

l. 95    Review use of “respectively” Sentence does not clarify which districts belong to which province?

l. 1787 – 80   Additional socio-demographic information on participants is useful here but sentence structure is awkward (remove repetitions?)

l. 590 – 594   Sentence must absolutely be reviewed as it is self-contradictory

Author Response

REVIEWER 1 COMMENTS – ROUND 2

Reviewer Comment 1: Thank you for this much improved version of your manuscript. We appreciate your thoughtfulness in answering our initial suggestions. 

Response: Thank you for all your feedback. We truly appreciate it.

Reviewer Comment 2: Overall, the Discussion section still needs to be edited as it too long and repeats many of the findings presented under results.

Response: We have made attempts to reduce the length of the discussion. However, we feel that it captures and conveys the intended information.

Reviewer Comment 3: The Limitations section needs to be expanded as the authors simply copied and pasted (verbatim) one of the suggestions made by the reviewers but the study has additional limitations in terms of validity and generalizability.

Response: We have expanded on the limitations of the study.

Reviewer Comment 4: Throughout the revised manuscript, the authors have made a diligent effort to clarify “influencers” vs “motivators”, but now need to be more rigorous in looking at “intrapersonal” vs “interpersonal”.

Response: Having reviewed the “intrapersonal” vs “interpersonal” in both the results and discussion, we feel that they align with the earlier clarification provided. We ask that the reviewer provides clarity on what we should focus on in addressing this comment.

Reviewer Comment 5: We have also noted a smattering of grammar and syntax errors or confusing statements that we think should be addressed before this manuscript can be published. We are listing a sample of them below, but we suggest that the Editorial Board give a thorough review and proofing of the manuscript before its publication.

Response: We have made efforts to address the grammar and syntax errors raised by the reviewers.

Abstract

Reviewer Comment 6: l. 12 Review sentence construction as it currently reads as if “low contraceptive prevalence prevents […] unsafe abortion”

Response: The sentence has been revised.

Main manuscript

Reviewer Comment 7: Review first sentence. It appears too broad and it’s unclear that adolescents globally bear a heavier burden of disease (than say, children under 5). Perhaps refocus on adolescent health issues?

Response: The sentence has been refocused

Reviewer Comment 8: l. 58    Review “low sexual debut”

Response: This has been revised.

Reviewer Comment 9: l. 95    Review use of “respectively” Sentence does not clarify which districts belong to which province?

Response: This sentence has been revised to improve clarity.

Reviewer Comment 10: l. 1787 – 80   Additional socio-demographic information on participants is useful here but sentence structure is awkward (remove repetitions?)

Response: The line numbering provided was not clear hence we were unable to address this specific comment.

Reviewer Comment 11: l. 590 – 594   Sentence must absolutely be reviewed as it is self-contradictory

Response: This sentence has been revised.
